# The Impact of Air Pollution on the Protection of World Cultural Heritage in China

**DOI:** 10.3390/ijerph191610226

**Published:** 2022-08-17

**Authors:** Bo Xiao, Lanyu Ning, Zixiang Lin, Shiyi Wang, Hua Zang

**Affiliations:** 1National Institute of Cultural Development, Wuhan University, Wuhan 430072, China; 2Institute of Quality Development Strategy, Wuhan University, Wuhan 430072, China; 3School of Economics and Business Foreign Languages, Wuhan Technology and Business University, Wuhan 430065, China; 4Department of Arts, Wuhan University, Wuhan 430072, China

**Keywords:** air pollution, world cultural heritage, sustainable protection, econometrics

## Abstract

The damage of air pollution to cultural heritage is widely known. However, the quantitative effects still need to be explored at a holistic level. Different from existing research which focuses on the “model calculation” methods, this paper uses an econometric approach to assess the overall impact of air pollution on the sustainable protection of world cultural heritage in China. Based on the data of the annual monitoring report from 2014 to 2020 released by the China World Cultural Heritage Monitoring Platform, this paper uses the thermal inversion as an instrument variable of air pollution to estimate the overall effects of air pollution on all world cultural heritage sites in China. The results indicate that almost all the air pollutants (except for CO) have significantly negative effects on heritage. The damaging effects of gaseous pollutants including SO_2_, NO_2_ and O_3_ is greater than that of particulate pollutants such as PM2.5 and PM10. Rainfall can exacerbate the worsening effects of gaseous pollutants, but will mitigate the negative effects of particulate pollutants; the windy weather may weaken the negative impact. In addition, environmental regulations from the local government can also alleviate the negative influence of air pollution on heritage protection. This research can provide a more comprehensive environmental prevention policy reference for the protection of world cultural heritage.

## 1. Introduction

Cultural heritage is an important carrier and integral component of human civilization, which intuitively reflects the important process of human social development. Cultural heritage is also indispensable physical evidence for the sustainable development of society. The protection of cultural heritage is the foundation for cultivating social and cultural development. As one of the birthplaces of world civilization, China is also one of the countries with the largest number of world cultural heritage sites. By 2020, China has successfully declared and been approved 55 world heritage sites. The Chinese government has been sparing no effort to protect the world cultural heritage.

Air pollution, which mainly includes gaseous pollutants and small solid pollutants, is one of the greatest threats to the sustainable protection of cultural heritage. The acid gases and oxidizing gases contained in air pollutants cause serious corrosion effects on stone, wood, metals, paints and other materials on the surface of the heritage through acidification reactions and oxidation reactions [1]. Sulfur oxides in the air are easily dissolved in water to generate sulfuric acid, which has a serious corrosion effect on metals, stones, murals and wood products [2]. Nitrogen dioxide can also be soluble in water and produce nitric acid, which has a strong corrosive effect on various materials of cultural monuments [3]. Ozone is a very strong oxidant, which only needs to be exposed to sunlight to oxidize materials such as wood and metal, accelerating its decay and rust [4]. Fine particulate matter in air pollutants (such as PM10, etc.) contains certain acidic and oxidizing substances, and will also adhere to the surface of the heritage material, causing wear and fouling [5]. Solid particles in the air tend to be irregular in shape and often angular. When they settle on the surface of cultural monuments with the flow of air, they will have a friction effect with them, resulting in certain mechanical wear on the surface of cultural relics [6]. In addition, particulate matters can also carry a large number of mold spores, insect eggs which may breed on a large scale under suitable conditions, thus causing serious damage to cultural relics [7].

The majority of the existing research has focused on assessing the impact of air pollution on cultural heritage through “model calculation” in the field of environmental science [1,8,9]. These “calculations” mainly rely on different models, including material corrosion models, air pollution distribution models, scene simulation models, and so on [5]. Models based on the participation of experiments and monitoring indicators can obtain more accurate quantitative results. This approach plays an important and positive role in the sustainable conservation of cultural heritage in specific locations or small areas (e.g., within a city). In many regions, corresponding preventive and protective measures for world cultural heritage are designed based on the results of accurate model calculations by environmental scientists [10]. However, if we want to assess the “overall” impact of air pollution on cultural heritage on a larger scale, the modelling approach to assessment is less applicable. This is mainly because the expansion of the scope of the assessment means that the uncertainty factor increases, and the accuracy and reliability of the model are challenged [9].

Unlike existing research, which focuses on “model calculation”, in this study, researchers use econometric methods to assess the “overall” impact of air pollution on the protection of world heritage in China through historical data. The research aims to provide a more comprehensive policy reference for the formulation of policies. To achieve this goal, there are two issues that need to be addressed. The first is to obtain time series data on the protection of cultural heritage. In 2013, the Chinese government established an online monitoring system for world cultural heritage. Since 2014, the government has published an annual monitoring report on the conservation status of each heritage site. The content of the report includes the degree of heritage integrity, the overall pattern change, the change of factor forms and materials, the change of disease, the change of land use, etc. We can use these reports to analyze the protection status of world cultural heritage in China over the years. The second is to address the possible endogenous problems between air pollution and the protection of cultural heritage. For example, on the one hand, regional economic development is more related to air pollution caused by regional industrial production. On the other hand, it will also affect the investment of funds for cultural heritage protection. We choose to use the thermal inversion as an instrument variable for air pollution to solve possible endogenous problems. As a meteorological phenomenon that occurs in the upper reaches of the atmosphere, it is weakly related to surface economic and social activities [11], and the emergence of inversion temperature will be unfavorable to surface air flow and aggravate air pollution [12].

Compared with the existing research, this paper mainly has the following two aspects of contribution. First, this paper uses econometric methods rather than environmental model calculations to assess the impact of air pollution on cultural monuments. Existing studies on assessing the impact of air pollution on cultural monuments focus on environmental science [13], mainly through corrosion experiments of air pollutants [14], monitoring and simulation of the distribution of atmospheric pollutants [1]. Few researchers use statistical methods from the perspective of economics. Based on the long-term time series data of world cultural heritage protection in China, this paper uses instrumental variable estimation method to evaluate the impact of air pollution on the protection of world cultural heritage in China. Second, this paper assesses the “holistic” effects of air pollution affecting cultural heritage on a national scale rather than a smaller site or region. Existing studies based on environmental model calculation methods have quantitatively assessed the local impact effects of air pollution on cultural heritage based on specific cultural monument sites or key areas [9,15]. Fewer studies can reveal the “holistic” impact of air pollution on cultural heritage protection on a larger scale. This paper uses econometric methods to assess the overall impact of air pollution on cultural heritage protection, which helps to provide a more comprehensive reference for cultural heritage protection policies at the national level.

## 2. Methodology

### 2.1. Models

The aim of this paper is to study the impact of air pollution on the protection of world cultural heritage in China. We follow the estimating method developed by Shi and Yu (2020) [16] and Shi et al. (2022) [17] and set the following Ordinary Least Square (OLS) model:(1)$$CH_{it}=\alpha_{0}+\alpha_{1}AP_{it}+\alpha_{2}X_{it}+\gamma_{i}+\lambda_{t}+\varepsilon_{it}$$

In Formula (1), $CH_{it}$ is the preservation status of the cultural heritage of the area i in the t year. $AP_{it}$ is the air pollution suffered by cultural heritage site i in t year. $X_{it}$ are other control variables related to cultural heritage protection and regional air pollution, mainly including some meteorological factors affecting regional air pollution, as well as the management strength and management level of local managerial departments affecting cultural heritage protection. $\gamma_{i}$ and $\lambda_{t}$ are site fixed effects and year fixed effects, and $\varepsilon_{it}$ is the error term. $\alpha_{1}$ is an estimated factor of interest to us, which represents the effects of air pollution on the cultural heritage.

In the model (1), we controlled the region and time, forming a two-way fixed effect, and also limiting the relevant meteorological conditions, the level of management of heritage protection and other factors. However, there are still other unobservable factors that may be related to both cultural heritage protection and regional air pollution that will have an impact on the estimation of model (1). For example, regional economic development is more related to air pollution caused by regional industrial production on the one hand, and on the other hand, it will also affect the investment of funds for cultural heritage protection. Therefore, the model (1) may have endogenous problems such as missing variables, which can lead to bias in the estimation results.

In order to solve the possible endogenous problems in model (1), this paper uses the “thermal inversion” as an instrument variable method for air pollution. Normally, the temperature in the atmospheric troposphere decreases with increasing altitude. This atmospheric junction is prone to convective movements, which can spread pollutants near the ground layer to the high and far reaches, thereby reducing the degree of air pollution in the city. However, under certain weather conditions, the atmospheric structure will have an anomalous phenomenon of increasing temperature with altitude, resulting in the stabilization of the atmospheric formation, which is called “thermal inversions”. The occurrence of inversion is not conducive to the upward movement of air. Thermal inversions make low-level water vapor and pollutants unable to diffuse to the upper air, resulting in the accumulation of air pollutants. The emergence of inversion will directly lead to an increase in air pollution. As a naturally occurring meteorological phenomenon, many studies have shown that its formation is not affected by regional economic or social factors. Therefore, it is also widely used in studies as an instrument variable for regional air pollution [18,19,20].

In this paper, we select the thermal inversion as an instrument variable for air pollution, and follow the Instrument Variable (IV) model developed by Shi and Yu (2020) [16] and Shi et al. (2022) [17] and use the two-stage least squares method (2SLS) to deal with the endogenous problem between air pollution and regional cultural heritage protection.

Stage 1:

(2)
$$AP_{it}=\beta_{0}+\beta_{1}TI_{it}+\beta_{2}X_{it}+\gamma_{i}+\lambda_{t}+\varepsilon_{it}$$

Stage 2:

(3)
$$CH_{it}=\alpha_{0}+\alpha_{1}\bar{AP_{it}}+\alpha_{2}X_{it}+\gamma_{i}+\lambda_{t}+\varepsilon_{it}$$



In Equation (2), the explanatory variable is the air pollution of the cultural heritage site, while the explanatory variable is the annual number of inversion days $TI_{it}$ of the site. Equation (3) replaces the original air pollution variable $\bar{AP_{it}}$ with the fitted air pollution variable $\bar{AP_{it}}$ in Equation (2). Since $\bar{AP_{it}}$ is determined by instrumental variables and other exogenous factors, the fitting value coefficient of $\bar{AP_{it}}$ can be regarded as unbiased. Other variables in Equations (2) and (3) are the same as those in Equation (1).

### 2.2. Data

#### 2.2.1. Evaluation of the Protection Condition of World Cultural Heritage

The evaluation data of the conservation status of world cultural heritage in this paper comes from the China World Cultural Heritage Monitoring Platform. In order to carry out unified monitoring of world cultural heritage, the Chinese government established an online monitoring system for world cultural heritage in 2013. The system is able to monitor the protection of cultural heritage in real time and provide tips and early warnings for poorly protected heritage. Since 2014, the China World Cultural Heritage Monitoring Platform has issued an annual monitoring report every year. The monitoring report assesses the protection of the heritage based on its location. The content of the assessment includes the degree of heritage integrity, the overall pattern change, the change of factor forms and materials, the change of disease, the change of land use, etc. Each metric is measured on the Likert scale (very good = 5, good = 4, general = 3, poor = 2, very poor = 1). Using these assessments in the annual report, this paper uses an arithmetic weighted average method to calculate the degree of cultural heritage protection obtained for each heritage from 2014 to 2020.

#### 2.2.2. Air Pollution

The air pollution data in this article comes from the China Air Quality History Data Set. The database platform collects the daily air pollutant index data of municipalities in China since 2013, mainly including the concentration values of PM2.5, PM10, NO_2_, SO_2_, CO, O_3_, etc. We matched each Prefecture-level city where the World Heritage Site is located, and obtained annual average concentration data for the air pollution indicators of each heritage site from 2014 to 2020.

#### 2.2.3. Thermal Inversions

This paper obtains the data of thermal inversions from space remote sensing information released by the National Aeronautics and Space Administration (NASA). NASA divides the Earth into a grid of latitude and longitude 0.5° × 0.625° and reports temperatures at 42 different sea levels every six hours since 1980. We collected the daily data for three sea levels closest to the surface from 2014 to 2020. First, we averaged the sea level temperature per day for each grid and counted the first layer below the second layer as 1 inversion day. Then, we added up the total number of inversion days per natural year for each grid. Finally, according to the Inverse Distance Weighted algorithm (IDW), the number and weight of each cultural heritage site in China were counted, and the number of inversion days per year for each cultural heritage from 2014 to 2020 was calculated.

#### 2.2.4. Control Variables

(1) Weather factors. On the one hand, meteorological factors will affect the formation and spread of air pollution; such as windy weather accelerating the dissipation of smog. On the other hand, climatic factors have a direct impact on the protection of cultural heritage, such as strong light or continuous rainfall may adversely affect woody monuments. Therefore, it is necessary to control weather factors. This article considers the control of the average annual temperature, average length of light, average humidity, precipitation days, and windy days in the sites of cultural heritage. Meteorological data are obtained from the China Meteorological Data Service Centre from 2014 to 2020.

(2) The effectiveness of cultural heritage protection. The management of the heritage by the local authorities may vary considerably from year to year, which will also have a significant impact on the results of our estimates. In this paper, we consider the control of the daily management of local heritage sites, the negative impact of tourists, and the investment in protection and remediation, so as to minimize the interference of human factors on the impact of air pollution on heritage protection in various places. Data on the day-to-day management of each heritage site, the negative impact of tourists, and the investment in conservation and remediation are all derived from the annual monitoring report on world heritage sites in China from 2014 to 2020.

The descriptive statistics for each variable are shown in Table 1.

## 3. Results

### 3.1. Results of OLS Model

First of all, we used model (1) to conduct OLS estimates of the effects of air pollution on heritage protection. The results are shown in Table 2. In all estimations, we controlled the weather factors including average temperature, relative humidity, precipitation, length of sunshine, rain and windy days, as well as the heritage site management factors such as the daily management, the impact of tourists and the renovation of heritage.

The results show that except for the pollutant CO, which has no significant impact on heritage protection, other pollutants have a significant negative impact on the preservation status of heritage. At the same time, from the perspective of the absolute value and significance of the estimated coefficient of each pollutant, the negative impact of gas pollutants (SO_2_, NO_2_ and O_3_) is greater than that of fine particulate matter (PM2.5 and PM10) pollutants. This may be because SO_2_ and NO_2_ are acidic gases that are corrosive to stony and woody remains, and O_3_ is a gas with strong oxidative properties. Together with these contaminants, we also found that PM10 became no longer significant. This suggests that fine particulate matters are mainly affected by PM2.5. The fine particles of PM10 are usually visible to the naked eye, so it is easier for us to take management measures such as cleaning or wiping the surface of the heritage. PM2.5 is more subtle and difficult to detect. As a result, PM2.5 can seep into the fine cracks of the heritage, causing it to develop lesions.

### 3.2. Results of IV Model

Since the model (1) fails to address the endogenous problem between air pollution and cultural heritage protection, it may lead to some bias in the estimation. This section uses the instrument variable regression method for further estimation, and the estimation results of various types of air pollutants are shown in Table 3.

Table 3 lists the estimation results of the first and second stages of various pollutants. From the first phase of the estimation, the estimation of thermal inversions for various types of pollutants is positive, and all of them are significant at the 1% level. This shows that the occurrence of thermal inversions does exacerbate the aggregation of various air pollutants. As an instrument variable for air pollution, it satisfies the strong correlation condition. From the estimated results of the second phase, the estimated coefficients of all types of pollutants except CO are significantly negative at the 1% level. This suggests that an increase in the concentration of these pollutants can significantly worsen the preservation of cultural heritage. Specifically, the annual average concentrations of PM2.5, PM10, SO_2_, NO_2_, and O_3_ in regional air pollutants increased by 1 unit per unit. This will result in an average reduction in the degree of preservation of cultural heritage by about 1.05%, 1.02%, 1.48%, 1.40% and 1.38%, respectively. Similarly, the negative impact of gaseous pollutants on the protection of cultural heritage is greater. The results of the instrument variable regression are basically consistent with the results of OLS regression, further validating the conclusion that air pollution can exacerbate the deterioration of cultural heritage.

At the same time, we further tested the effectiveness of the thermal inversion as an IV. In Table 3, the Kleibergen-Paap unrecognized test is carried out for each two-stage estimation model. The test results show that LMstatistics is at the significant indigenous level, excluding the unrecognized hypothesis. In addition, we conducted Cragg-Donald and Kleibergen-Paap Wald tests on thermal inversions, respectively. The results showed that the F statistic was much larger than 10, and all of them were significant at 1% level. Therefore, we also exclude the possibility of weak instrumental variables. It is reasonable and effective to choose thermal inversion as an instrument variable for air pollution.

### 3.3. Robustness Tests

In order to ensure the reliability of the conclusions of this study, we further tested the robustness of the estimated results. We selected PM2.5 of solid fine particles and SO_2_ of gaseous pollutants as representatives of major air pollution for testing. The study still uses the instrumental variable method for estimation. Table 4 and Table 5 report the second-stage estimation results of PM2.5 and SO_2_.

First of all, we considered that between 2014 and 2020, the Chinese government may introduce corresponding cultural heritage protection policies. These policies will have a direct impact on the extent and manner in which cultural heritage is protected everywhere, which in turn will have an impact on our previous estimates. Therefore, we examined the impact of the ‘Guidance on Further Strengthening Cultural Relics Work‘ promulgated by the Chinese government in 2016, and the newly revised ‘Cultural Relics Protection Law’ in 2017 on the estimation results. We estimated this by setting the time dummy variable before and after the policy and adding it as a new control variable to the regression equation. The results are shown in columns (1) and (2) of Table 4 and Table 5. Compared with the estimated results of PM2.5 and SO_2_ in Table 3, there is no significant change in the size and significance of the coefficients. The implementation of cultural heritage protection policies has not affected the results of this research.

Next, we considered the relevant environmental protection policies issued by the Chinese government in the sample range. These policies will have a direct impact on air pollution in cultural heritage areas and thus may also affect our estimates. We examined the new environmental protection law implemented in 2015, the central ecological and environmental protection inspectorate between 2016 and 2019, and the ecological civilization demonstration zone that began in 2014. For the new environmental protection law of 2015, we estimate it by setting a time dummy variable before and after the policy. For the central ecological environmental protection inspector and the policy of the ecological civilization demonstration area, we estimate it by setting a virtual variable of whether the municipality where the cultural heritage is located was affected by the policy in that year. The estimated results are as shown in the columns (3), (4) and (5) in Table 4 and Table 5, respectively. As can be seen from the results, none of the estimated coefficients have changed significantly, indicating that environmental regulatory policies have not significantly affected the previous estimates in this paper.

Then, considering the impact of extreme values in the sample on the estimation results, we deleted some special samples to test. Large economically developed cities will be able to invest more money or technology in the protection of the cultural heritage of their locations, which could seriously weaken the negative effect of air pollution on cultural heritage. Therefore, we deleted samples from the samples belonging to the two megacities of Beijing and Shanghai. The results are shown in column (6) of Table 4 and Table 5. The negative effects of air pollution on the preservation of cultural heritage may only exist in some areas with very high levels of pollution, while in other areas the effect may not be obvious. This also inflicts biases on the previous estimates, so we removed samples of air pollutant PM2.5 and SO_2_ extreme concentrations (based on concentration values above 90% and below 10%). The results are shown in column (7) of Table 4 and Table 5. Similarly, we found no significant change in the results compared to the results corresponding to the previous Table 3.

Finally, we took a more stringent ‘time-space interaction fixed effect’ control into account. Since Chinese culture usually has strong inter-provincial differences, we controlled the fixed effect of the interaction between the province where the cultural heritage is located and the year, so as to strip away the factors related to the change of the province where the cultural heritage is located over time. The results are shown in column (8) of Table 4 and Table 5. Although the absolute value of the estimated coefficient of SO_2_ decreased slightly, the estimation coefficient is still negatively significant, indicating that the previous results are robust.

## 4. The Effects of Moderating Factors

This section explores some of the regulatory elements, including weather factors, the intensity of environmental regulation of heritage sites, and the role of air pollution in worsening the sustainable protection of cultural heritage. By considering these factors, research can reveal the impact of air pollution on the sustainable protection of cultural heritage more comprehensively.

### 4.1. Moderating Effects of Weather

#### 4.1.1. The Moderate Effects of Rain

Rainfall can significantly affect the extent of air pollution. It is generally believed that the washing of rainwater can largely remove all kinds of pollutants floating in the air, thereby slowing down the damage of air pollution to items. We captured the regulatory role of rainfall by interacting with the number of days of annual rainfall in cultural heritage locations with pollutant concentrations. The results are shown in Table 6. As can be seen from the results, columns (1) and (2) show positive and significant results. Columns (3) and (4) are negative. Columns (5) and (6) are not significant, indicating that the regulatory effect of rainfall on different air pollutants is significantly various. Specifically, rainfall significantly exacerbates the negative impact of acid gas pollutants (SO_2_ and NO_2_) on heritage protection. However, rainfall reduces the negative impact of solid fine particulate pollutants (PM2.5 and PM10) on heritage conservation, while also mitigating the effects of neutral gases (O_3_). It is well known that acid gases such as SO_2_ and NO_2_ can dissolve in rainwater through rain to form liquids that can cause corrosiveness to wood, metal or slate. Thus, rainfall exacerbates the exacerbating effect of such gas pollution on cultural heritage. But for fine-grained contaminants such as PM2.5 and PM10, the erosion of rainwater can clean up contaminants that were originally attached to the surface of the item, protecting it from the risk of destruction. This reduces its negative impact on cultural heritage.

#### 4.1.2. The Moderate Effects of Wind

Similarly, winds can accelerate the spread of various types of air pollutants. The concentration of pollutants is significantly reduced, thereby reducing the damage of air pollution to items. In this study, the annual wind blowing days in the heritage site interacted with the concentrations of pollutants to estimate the moderating effect of wind blowing weather; the results are shown in Table 7. As can be seen from the regression results, the estimated coefficients for each interaction term are significantly positive (except CO), indicating that winds significantly reduce the negative impact of gases and solid fine particulate pollutants on heritage conservation. As a result, windy weather reduces regional air pollution levels, thereby weakening the worsening effects of air pollution on cultural heritage. Accelerating air movement near cultural heritage sites could also be a coping strategy to mitigate the negative effects of air pollution.

### 4.2. Moderating Effects of Regional Environmental Regulation

The effectiveness of regional environmental regulations directly determines the level of pollution emissions, thus having a significant impact on regional air pollution. In this section, we examined the moderating effect of environmental regulation effectiveness in heritage sites. According to Yu et al., (2021, 2022) [21,22], we measured the intensity of government environmental regulation by the frequency of environmental words in the annual government work report [23] of the prefecture-level city where the heritage resides. Similarly, regression estimation is performed by adding interactive terms, and the results are shown in Table 8. The interaction coefficient between the effectiveness of regional environmental regulations and PM2.5, PM10 and SO_2_ is significantly positive, indicating that the intensity of regional environmental regulations significantly reduces the aggravating effect of air pollution on cultural heritage. However, the interaction between regional environmental regulation effectiveness and NO_2_ and O_3_ is not significant, especially the interaction term coefficient with O_3_ is still negative. This may be related to pollutants such as PM2.5, PM10 and SO_2_, which are not the focus of regional environmental regulations. Overall, we believe that local government environmental regulations can help significantly reduce the negative impact of gases and solid fine particulate pollutants on heritage conservation. Strengthening local governments’ efforts to control air pollution can also indirectly play a positive role in protecting cultural heritage.

## 5. Conclusions and Discussion

People are aware of the damage of air pollution on cultural heritage sites, but they have not explored its quantified effects at a holistic level. Existing studies focus on quantitatively assessing the impact effects of specific cultural heritage sites and small regions based on “model calculations”. This paper uses econometric inference to assess the impact of air pollution on the sustainable protection of world heritage in China.

Based on the data of the 2014–2020 annual monitoring report released by the China World Cultural Heritage Monitoring and Early Warning Platform, this paper selects the thermal inversion as an instrument variable for air pollution to deal with the endogenous problem between air pollution and regional cultural heritage protection. The study uses a two-stage least squares method to estimate the impact of air pollution on cultural heritage conservation nationwide. The results show that except for the pollutant CO, which has no significant impact on heritage protection, other pollutants have a significant negative impact on the preservation status of heritage. Compared with fine particulate matter (PM2.5 and PM10) pollutants, gaseous pollutants (SO_2_, NO_2_ and O_3_) have a greater negative effect. This may be related to the corrosiveness of SO_2_ and NO_2_ as acid gases to stony and woody remains and the strong oxidation properties of O_3_. A series of robustness tests have also confirmed these findings. We also discussed the regulatory effect of meteorological factors. Rainfall exacerbates the worsening effects of gaseous pollutants on cultural heritage, but will reduce the negative effects of solid fine particulate pollutants on cultural heritage. Winds reduce regional air pollution levels, thereby weakening the exacerbating effect of air pollution on cultural heritage. In addition, local government environmental regulations help significantly reduce the negative impact of gases and solid fine particulate pollutants on heritage conservation. Strengthening local governments’ efforts to control air pollution can indirectly play a positive role in protecting cultural heritage.

In view of the above, on the protection of cultural heritage, the government of the heritage site should put the concern and prevention of air pollution on the agenda, and take a series of measures to effectively deal with the negative impact. Firstly, the government should strengthen the emphasis and coordination of environmental protection at the site level, incorporate environmental protection work and heritage protection into development planning, and comprehensively consider the positive impact of the two. Secondly, at the level of policies and regulations, the government needs to properly handle the relationship between environmental protection and economic and social development, and establish a clear special system and local regulations to protect the world cultural heritage from the impact of air pollution. Thirdly, at the micro level, the government should explore an optimal path to reduce air pollution and protect cultural heritage through a series of measures such as civil air defense, material defense and technical defense. At the same time, the national environmental protection department and the cultural relics department should strengthen cooperation and jointly explore environmental protection plans within and around the control of cultural heritage. The national sectors should give the necessary consideration and support in terms of macro policies and financial guarantees. In addition, it is necessary for the government to promote the popularization of cultural heritage environmental protection policies and increase the enthusiasm of communities, enterprises and residents in cultural heritage sites. Ultimately, in order to improve the effectiveness of the world cultural heritage protection continually, the government has to take the responsibility to mobilize a wide range of social forces to jointly devote themselves to the cause of cultural heritage protection.

## Figures and Tables

**Table 1 ijerph-19-10226-t001:** Descriptive statistics of variables.

Variables	Sample Size	Mean Value	Standard Deviation	Min	Max
Evaluation of World Cultural Heritage Protection (EWCHP)	520	3.436	0.658	0.200	5
PM2.5 (µg/m^3^)	520	48.368	38.684	34.562	214.642
PM10 (µg/m^3^)	520	86.734	58.793	45.454	523.654
SO_2_ (µg/m^3^)	520	26.984	22.842	8.453	136.975
NO_2_ (µg/m^3^)	520	30.574	16.634	11.643	108.464
CO (mg/m^3^)	520	1.003	0.597	0.103	8.254
O_3_ (µg/m^3^)	520	56.987	28.422	14.544	98.452
Thermal inversion (days)	520	72.561	32.201	8.563	112.343
Average temperature (°C)	520	13.983	4.792	−2.989	29.342
Relative humidity	520	62.542	6.153	26.432	83.023
Precipitation	520	1073.433	545.663	31.353	3342.324
Sunshine duration	520	2105.053	411.451	653.146	3137.534
Rainy days	520	89.353	15.636	23.636	135.633
Windy days	520	93.673	19.637	35.473	253.745
Daily management (very good = 5; good = 4; general = 3; poor = 2; very poor = 1)	520	3.856	1.543	1	5
Negative effects of tourists (If the heritage site has been negatively affected by tourists, enter 1; Instead, enter 0)	520	0.142	0.0243	0	1
Protection and renovation (very good = 5; good = 4; general = 3; poor = 2; very poor = 1)	520	3.298	1.283	1	5

**Table 2 ijerph-19-10226-t002:** OLS Estimations of the impact of air pollution on cultural heritage protection.

Explained Variable: EWCHP	(1)	(2)	(3)	(4)	(5)	(6)	(7)
PM2.5	−0.018 **						−0.016 **
	(0.009)						(0.008)
PM10		−0.019 **					−0.010
		(0.009)					(0.009)
SO_2_			−0.028 ***				−0.020 ***
			(0.008)				(0.006)
NO_2_				−0.022 **			−0.018 **
				(0.008)			(0.005)
CO					−0.012		−0.001
					(0.009)		(0.006)
O_3_						−0.024 ***	−0.017 ***
						(0.008)	(0.006)
Weather controls	Y	Y	Y	Y	Y	Y	Y
Management controls	Y	Y	Y	Y	Y	Y	Y
Site fixed effects	Y	Y	Y	Y	Y	Y	Y
Year fixed effects	Y	Y	Y	Y	Y	Y	Y
N	520	520	520	520	520	520	520
R^2^	0.602	0.618	0.619	0.589	0.596	0.624	0.689

** and *** indicate that the significance levels at 5%, and 10%, respectively; Standard errors are in parentheses; Weather controls include average temperature, relative humidity, precipitation, sunshine duration, rainfall and wind days, Y means those variables have been controlled in estimations; Management controls include daily management of the heritage, the impact of tourists and the effectiveness of protection and renovation. Y means those variables have been controlled in estimations.

**Table 3 ijerph-19-10226-t003:** IV estimation of the impact of air pollution on cultural heritage protection.

	(1)		(2)		(3)		(4)		(5)		(6)	
	Stage 1	Stage 2	Stage 1	Stage 2	Stage 1	Stage 2	Stage 1	Stage 2	Stage 1	Stage 2	Stage 1	Stage 2
Explained Variables	PM2.5	EWCHP	PM10	EWCHP	SO_2_	EWCHP	NO_2_	EWCHP	CO	EWCHP	O_3_	EWCHP
PM2.5		−0.036 ***										
		(0.002)										
PM10				−0.035 ***								
				(0.003)								
SO_2_						−0.051 ***						
						(0.004)						
NO_2_								−0.048 ***				
								(0.003)				
CO										−0.011		
										(0.010)		
O_3_												−0.047 ***
												(0.003)
Thermal inversion	0.135 ***		0.124 ***		0.120 ***		0.110 ***		0.090 ***		0.110 ***	
	(0.011)		(0.010)		(0.009)		(0.007)		(0.014)		(0.010)	
Weather controls	Y	Y	Y	Y	Y	Y	Y	Y	Y	Y	Y	Y
Management controls	Y	Y	Y	Y	Y	Y	Y	Y	Y	Y	Y	Y
Site fixed effects	Y	Y	Y	Y	Y	Y	Y	Y	Y	Y	Y	Y
Year fixed effects	Y	Y	Y	Y	Y	Y	Y	Y	Y	Y	Y	Y
KP LM statistic		274.644 ***		274.644 ***		212.846 ***		212.846 ***		212.846 ***		212.846 ***
CD Wald F statistic		326.295 ***		326.295 ***		247.086 ***		247.086 ***		247.086 ***		247.086 ***
KP Wald F statistic		279.271 ***		279.271 ***		214.983 ***		214.983 ***		214.983 ***		214.983 ***
N	520	520	520	520	520	520	520	520	520	520	520	520
R^2^		0.667		0.632		0.605		0.619		0.614		0.629

*** indicates that the significance levels at 10%; Standard errors are in parentheses; Weather controls include average temperature, relative humidity, precipitation, sunshine duration, rainfall and wind days, Y means those variables have been controlled in estimations; Management controls include daily management of the heritage, the impact of tourists and the effectiveness of protection and renovation. Y means those variables have been controlled in estimations.

**Table 4 ijerph-19-10226-t004:** Robustness test of PM2.5 estimates.

Explained Variable: EWCHP	(1)	(2)	(3)	(4)	(5)	(6)	(7)	(8)
PM2.5	0.034 ***	0.034 ***	0.033 ***	0.037 ***	0.036 ***	0.038 ***	0.030 ***	0.038 ***
	(0.002)	(0.003)	(0.002)	(0.003)	(0.003)	(0.003)	(0.003)	(0.003)
Weather controls	Y	Y	Y	Y	Y	Y	Y	Y
Management controls	Y	Y	Y	Y	Y	Y	Y	Y
Site fixed effects	Y	Y	Y	Y	Y	Y	Y	Y
Year fixed effects	Y	Y	Y	Y	Y	Y	Y	N
The New Law of the Peoples Republic of China on Protection of Cultural Relics	Y							
Policies to strengthen the protection of cultural relics		Y						
The new law of environmental protection			Y					
Central environmental protection supervision				Y				
Ecological civilization demonstration area					Y			
Results after deleting the super city						Y		
Results after deleting pollution extremum							Y	
The interactive fixed effect of provinces and years								Y
N	520	520	520	520	520	364	420	520
R^2^	0.689	0.680	0.672	0.691	0.626	0.690	0.672	0.778

*** indicates that the significance levels at 10%; Standard errors are in parentheses; Weather controls include average temperature, relative humidity, precipitation, sunshine duration, rainfall and wind days, Y means those variables have been controlled in estimations; Management controls include daily management of the heritage, the impact of tourists and the effectiveness of protection and renovation. Y means those variables have been controlled in estimations.

**Table 5 ijerph-19-10226-t005:** Robustness test of SO_2_ estimation.

Explained Variable: EWCHP	(1)	(2)	(3)	(4)	(5)	(6)	(7)	(8)
SO_2_	0.044 ***	0.044 ***	0.043 ***	0.047 ***	0.046 ***	0.048 ***	0.042 ***	0.037 ***
	(0.004)	(0.004)	(0.005)	(0.005)	(0.005)	(0.005)	(0.004)	(0.004)
Weather controls	Y	Y	Y	Y	Y	Y	Y	Y
Management controls	Y	Y	Y	Y	Y	Y	Y	Y
Site fixed effects	Y	Y	Y	Y	Y	Y	Y	Y
Year fixed effects	Y	Y	Y	Y	Y	Y	Y	N
The New Law of the Peoples Republic of China on Protection of Cultural Relics	Y							
Policies to strengthen the protection of cultural relics		Y						
The new law of environmental protection			Y					
Central environmental protection supervision				Y				
Ecological civilization demonstration area					Y			
Results after deleting the super city						Y		
Results after deleting pollution extremum							Y	
The interactive fixed effect of provinces and years								Y
N	520	520	520	520	520	364	420	520
R^2^	0.667	0.686	0.657	0.657	0.653	0.635	0.667	0.764

*** indicates that the significance levels at 10%; Standard errors are in parentheses; Weather controls include average temperature, relative humidity, precipitation, sunshine duration, rainfall and wind days, Y means those variables have been controlled in estimations; Management controls include daily management of the heritage, the impact of tourists and the effectiveness of protection and renovation. Y means those variables have been controlled in estimations.

**Table 6 ijerph-19-10226-t006:** Estimation of adjustment effect of rainfall.

Explained Variable: EWCHP	(1)	(2)	(3)	(4)	(5)	(6)
PM2.5 * Rainy_days	0.005 **					
	(0.002)					
PM10 * Rainy_days		0.002 **				
		(0.001)				
SO_2_ * Rainy_days			−0.025 ***			
			(0.002)			
NO_2_ * Rainy_days				−0.022 ***		
				(0.006)		
CO * Rainy_days					0.005	
					(0.005)	
O_3_ * Rainy_days						0.004
						(0.005)
Air pollutants	Y	Y	Y	Y	Y	Y
Weather controls	Y	Y	Y	Y	Y	Y
Management controls	Y	Y	Y	Y	Y	Y
Site fixed effects	Y	Y	Y	Y	Y	Y
Year fixed effects	Y	Y	Y	Y	Y	Y
N	520	520	520	520	520	520
R^2^	0.612	0.624	0.629	0.603	0.606	0.632

** and *** indicate that the significance levels at 5%, and 10%, respectively; Standard errors are in parentheses; Weather controls include average temperature, relative humidity, precipitation, sunshine duration, rainfall and wind days, Y means those variables have been controlled in estimations; Management controls include daily management of the heritage, the impact of tourists and the effectiveness of protection and renovation. Y means those variables have been controlled in estimations. Air pollutants include concentrations of PM2.5, PM10, SO_2_, NO_2_, CO, and O_3_. Y means those variables have been controlled in estimations.

**Table 7 ijerph-19-10226-t007:** The moderate effects of wind.

Explained Variable: EWCHP	(1)	(2)	(3)	(4)	(5)	(6)
PM2.5 * Windy_days	0.008 ***					
	(0.002)					
PM10 * Windy_days		0.004 **				
		(0.002)				
SO_2_ * Windy_days			0.005 **			
			(0.002)			
NO_2_ * Windy_days				0.003 ***		
				(0.001)		
CO * Windy_days					0.003	
					(0.004)	
O_3_ * Windy_days						0.004 **
						(0.002)
Air pollutants	Y	Y	Y	Y	Y	Y
Weather controls	Y	Y	Y	Y	Y	Y
Management controls	Y	Y	Y	Y	Y	Y
Site fixed effects	Y	Y	Y	Y	Y	Y
Year fixed effects	Y	Y	Y	Y	Y	Y
N	520	520	520	520	520	520
R^2^	0.612	0.624	0.629	0.603	0.606	0.632

** and *** indicate that the significance levels at 5%, and 10%, respectively; Standard errors are in parentheses; Weather controls include average temperature, relative humidity, precipitation, sunshine duration, rainfall and wind days, Y means those variables have been controlled in estimations; Management controls include daily management of the heritage, the impact of tourists and the effectiveness of protection and renovation. Y means those variables have been controlled in estimations; Air pollutants include concentrations of PM2.5, PM10, SO_2_, NO_2_, CO, and O_3_. Y means those variables have been controlled in estimations.

**Table 8 ijerph-19-10226-t008:** Moderating effects of regional environmental regulation.

Explained Variable: EWCHP	(1)	(2)	(3)	(4)	(5)	(6)
PM2.5 * Intensity	0.007 **					
	(0.004)					
PM10 * Intensity		0.008 **				
		(0.004)				
SO_2_ * Intensity			0.005 *			
			(0.003)			
NO_2_ * Intensity				0.003		
				(0.003)		
CO * Intensity					−0.002	
					(0.005)	
O_3_ * Intensity						−0.007
						(0.005)
Air pollutants	Y	Y	Y	Y	Y	Y
Weather controls	Y	Y	Y	Y	Y	Y
Management controls	Y	Y	Y	Y	Y	Y
Site fixed effects	Y	Y	Y	Y	Y	Y
Year fixed effects	Y	Y	Y	Y	Y	Y
N	520	520	520	520	520	520
R^2^	0.646	0.673	0.687	0.646	0.656	0.635

* and ** indicate that the significance levels at 1%, 5%, respectively; Standard errors are in parentheses; Weather controls include average temperature, relative humidity, precipitation, sunshine duration, rainfall and wind days, Y means those variables have been controlled in estimations; Management controls include daily management of the heritage, the impact of tourists and the effectiveness of protection and renovation. Y means those variables have been controlled in estimations; Air pollutants include concentrations of PM2.5, PM10, SO_2_, NO_2_, CO, and O_3_. Y means those variables have been controlled in estimations.

## Data Availability

The data that support the findings of this study are available from the corresponding author upon reasonable request.

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
