# Peer review of "The Impact of Air Pollution on the Protection of World Cultural Heritage in China"

_ijerph, 2022, doi:10.3390/ijerph191610226_

Round 1

Reviewer 1 Report

Air pollution is an important factor in the damage to cultural heritage. The quantitative estimation of the damage of air pollution to cultural heritage is need. This paper uses econometric approach to assess the overall impact of air pollution on the sustainable protection of world cultural heritage in China. It’s helpful to assess the overall impact of air pollution on cultural heritage protection and further to provide a more comprehensive reference for cultural heritage protection policies at the national level. However, some detailed information were needed to make the result reliable.

1.      Major concerns.

(1)     The structure of this paper is need to improve. Such as: (1) Line 79-87 in “Introduction”, the content is similar with abstract. (2) “2. Literature review” is a separate section while this paper is not a review paper.

(2)     As for “3.1 Models”, the author present Formula (1) for model 1, Formula (2) and (3) for 2SLS and no reference cited. How to verify the rationality of the methods? Are these methods used in similar studies?

(3)     This study involves many factors, such as the EWCHP, air pollutants, thermal inversion, weather factors and so on. But there’s no such data in the paper besides a very rough description in Table 1.

(4)     As for the study period, it’s different in 3.2.1 and 3.2.2 while there’s no time period in 3.2.3 and 3.2.4.

(5)     As for the Table 2-Table 8, what’s the meaning of the number (or the column represented by the number) in the first line?

2.      Specific comments

(1)    The information in the tables is incomplete, such as the time period, data in parentheses, Y and so on.

(2)    The abbreviation should give full name when it appeared first time, for example, OLS, IV, and so on.

Author Response

Dear reviewer,

Thank you so much for your comments! Those comments are very helpful in improving our paper. According to your comments, we have substantially revised our manuscript. You can check it in our new manuscript.

Best regards,

Zang Hua

Reviewer 2 Report

Generally a robust paper - there may be some concerns as to how robust the regression is, considering tht exogenous factors may also play a role and need ot be controlled for, whether they are weather variables, rainfall, temperature, altitude etc, it would be interesting to see a control for these based on individual gas behaviors.

Overall, relatively good, coudl do with a general proof read and tweak of English usage.

Author Response

Dear reviewer:

Thank you so much for your comments! We have re-examined and perfected the English expression of the paper. You can check it in our new manuscript.

Best regards, 

Zang Hua

Round 2

Reviewer 1 Report

 The authors have revised the manuscript according the reviewers’ comments. There’s still some minor mistake that the author should read and revise carefully. For example, Line 204-205, “Meteorological data are obtained from the China Meteorological Data Service Centre2 from 2014 to 2020”. “2” may be deleted. 

Author Response

Dear reviewer:

Thank you for your comments! We have fully accepted your suggestion and carefully checked and corrected the minor errors in our manuscript, which you can read in our revised manuscript.

Thank you once again for your efforts on our paper.

Best,